# THE IMPACTS OF KNOWN AND UNKNOWN DEMONSTRATOR IRRATIONALITY ON REWARD INFERENCE

## ABSTRACT

Algorithms inferring rewards from human behavior typically assume that people are (approximately) rational. In reality, people exhibit a wide array of irrationalities. Motivated by understanding the benefits of modeling these irrationalities, we analyze the effects that demonstrator irrationality has on reward inference. We propose operationalizing several forms of irrationality in the language of MDPs, by altering the Bellman optimality equation, and use this framework to study how these alterations affect inference.

We find that incorrectly assuming noisy-rationality for an irrational demonstrator can lead to remarkably poor reward inference accuracy, even in situations where inference with the correct model leads to good inference. This suggests a need to either model irrationalities or find reward inference algorithms that are more robust to misspecification of the demonstrator model. Surprisingly, we find that if we give the learner access to the correct model of the demonstrator's irrationality, these irrationalities can actually *help* reward inference. In other words, if we could choose between a world where humans were perfectly rational and the current world where humans have systematic biases, the current world might counter-intuitively be preferable for reward inference. We reproduce this effect in several domains. While this finding is mainly conceptual, it is perhaps actionable as well: we might ask human demonstrators for myopic demonstrations instead of optimal ones, as they are more informative for the learner and might be easier for a human to generate.

## 1 INTRODUCTION

Motivated by difficulty in reward specification (Lehman et al., 2018), inverse reinforcement learning (IRL) methods estimate a reward function from human demonstrations (Ng et al., 2000; Abbeel and Ng, 2004; Kalman, 1964; Jameson and Kreindler, 1973; Mombaur et al., 2010). The central assumption behind these methods is that human behavior is rational, i.e., optimal with respect to their reward (cumulative, in expectation). Unfortunately, decades of research in behavioral economics and cognitive science (Chipman, 2014) have unearthed a deluge of *irrationalities*, i.e., of ways in which people deviate from optimal decision making: hyperbolic discounting, scope insensitivity, illusion of control, decision noise, loss aversion, to name a few.

While as a community we are starting to account for some possible irrationalities plaguing demonstrations in different ways (Ziebart et al., 2008; 2010; Singh et al., 2017; Reddy et al., 2018; Evans et al., 2016; Shah et al., 2019), we understand relatively little about what effect irrationalities have on the difficulty of inferring the reward. In this work, we seek a systematic analysis of this effect. Do irrationalities make it harder to infer the reward? Is it the case that the more irrational someone is, the harder it is to infer the reward? Do we need to account for the specific irrationality type during learning, or can we get away with the standard noisy-rationality model? The answers to these questions are important in deciding how to move forward with reward inference. If irrationality, even when well modelled, still makes reward inference very difficult, then we will need alternate ways to specify behaviors. If well-modelled irrationality leads to decent reward inference but we run into problems when we just make a noisy-rational assumption, that suggests we need to start accounting for irrationality more explicitly, or at least seek assumptions or models that are robust to many different types of biases people might present. Finally, if the noisy-rational model leads

Figure 1: We found that irrationality does not always hinder reward inference (section 3.2) – it is in many cases actually helpful. Here, we depict rational and myopic (short-sighted) behavior in a merging environment (section 4.2) for two different rewards, with higher and lower weights on going fast. The rational car (white) exhibits similar behavior under both reward functions, while the myopic car (blue) overtakes on the shoulder when the reward function places a high weight on speed. This makes it easier to differentiate what its reward is.

to decent inference even when the demonstrator is irrational, then we need not dedicate significant resources to addressing irrationality.

One challenge with conducting such an analysis is that there are many irrationalities in the psychology and behavioral economics literature, with varying degrees of mathematical formalization versus empirical description. To structure the space for our analysis, we operationalize irrationalities in the language of MDPs by systematically enumerating possible deviations from the Bellman equation – imperfect maximization, deviations from the true transition function, etc. This gives us a formal framework in which we can simulate irrational behavior, run reward inference, and study its performance. Armed with this formalism, we then explore the various impacts of irrationality on reward learning in three families of environments: small random MDPs, a more legible gridworld MDP, and an autonomous driving domain drawn from the robotics literature (Sadigh et al., 2016).

**Irrationality can help, rather than hinder reward inference – if it is modelled correctly.** We first explore the impacts of demonstrator irrationality when the irrationality is known to the reward inference algorithm. Surprisingly, we find that certain irrationalities actually *improve* the quality of reward inference - that is, they make reward easier to learn. Importantly, this is not compared to assuming the wrong model of the human: our finding is that humans who exhibit (correctly modelled) irrationality are more informative than humans who exhibit (correctly modelled) rationality! This is consistent in all three domains. We explain this theoretically from the perspective of the *mutual information* between the demonstrator behavior and the reward parameters, proving that some irrationalities are arbitrarily more informative than rational behavior.

**Unmodelled irrationality leads to remarkably poor reward inference.** It might seem that we can't immediately benefit from the knowledge that irrationalities help inference unless we have a comprehensive understanding of human decision-making, and so we should just stick to the status quo of modeling people as rational. However, we find that modeling irrational demonstrators as noisily-rational can lead to worse outcomes than not performing inference at all and just using the prior (section 5). Encouragingly, we also find evidence that even just modeling the demonstrator's irrationality approximately allows a learner to outperform modeling the demonstrator as noisily-rational (section E).

Overall, we contribute 1) a theoretical and empirical analysis of the effects of different irrationalities on reward inference, 2) a way to systematically formalize and cover the space of irrationalities in order to conduct such an analysis, and 3) evidence for the importance and benefit of accounting for irrationality irrationality during inference.

Our results suggest that modeling people as noisily rational leads to poor reward inference, and that it is important to model the irrationalities of human demonstrators. Our good news is that if we manage to do that well, we might be better off even compared to a counterfactual world in which people are actually rational! Of course, modeling irrationality is a long term endeavour. Our near-term good news is two fold: first, irrationalities can be an ally for teaching. For example, we could ask human demonstrators to act more myopically to better communicate their reward to the learners. Second, we need not get the biases perfectly correct to do better than assuming noisy-rationality. Instead, using slightly more realistic models of human irrationality could lead to better inference.

Myopic Discounting

Myopic Horizon

Boltzmann-Rationality

Prospect Bias

Hyperbolic Discounting

$$V_{i+1}(s) = \max_a \sum_{s' \in S} P_{s,a}(s') \left( r_\theta(s, a, s') + \gamma V_i(s') \right)$$

Optimism/Pessimism
Illusion of Control

Extremal Bias

Figure 2: In section 2.3, we modify the components of the Bellman update to cover different types of irrationalities: changing the max into a softmax to capture noise (Boltzmann), changing the transition function to capture optimism/pessimism or the illusion of control, changing the reward values to capture the nonlinear perception of gains and losses (Prospect), changing the average reward over time into a maximum (Extremal), and changing the discounting to capture more myopic decision-making.

## 2 FRAMEWORK: BIASES AS DEVIATIONS FROM THE BELLMAN UPDATE

### 2.1 EXPLORING BIASES THROUGH SIMULATION

While ideally we would recruit human subjects with different irrationalities and measure how well we can learn rewards, this is prohibitive because we do not get to dictate someone's irrationality type: people exhibit a mix of them, some yet to be discovered. Further, measuring the accuracy of inference from observing real humans is complicated by the fact that we do not have ground truth access to the human's reward function. For instance, suppose we asked subjects to produce a set of (behavior, reward function) pairs. We could then try to predict the reward functions from the behaviors. But how did we, the experimenters, infer the reward functions from the people? If we are wrong in our assumptions about which irrationalities are affecting their behavior and/or explicit reports of rewards, we would remain deluded about the subjects' true intentions and preferences.

To address these issues, we *simulate* demonstrator behavior subject to different irrationalities, run reward inference, and measure the performance against the ground truth, i.e., the accuracy of a Bayesian posterior on the reward parameter given the (simulated) demonstrator's inputs.

### 2.2 BACKGROUND AND FORMALISM

Consider an Uncertain-Reward MDP (URMDP) (Bagnell et al., 2001; Regan and Boutilier, 2011; Desai, 2017) $\mathcal{M} = (S, A, \{P_{s,a}\}, \gamma, \Theta, p, r)$, consisting of finite state and action sets $S$ and $A$, distributions over states $\{P_{s,a}\}$ representing the result of taking action $a$ in state $s$, discount rate $\gamma \in [0, 1)$, a (finite) set of reward parameters $\Theta$, a prior distribution $p \in \Delta(S \times \Theta)$ over starting states and reward parameters, and a parameterized state-action reward function $r : \Theta \times S \times A \times S \to \mathbb{R}$, where $r_\theta(s, a, s')$ represents the reward received.

We assume that the human demonstrator's policy $\pi$ satisfies $\pi = d(\theta)$, where $d$ is an (environment-specific) **planner** $d : \Theta \to \Pi$ that returns a (possibly stochastic) policy given a particular reward parameter $\theta$. The **rational** demonstrator uses a planner $d_{\text{Rational}}$ that, given a reward parameter $\theta$, returns a policy that maximizes its expected value. On the other hand, we say that an demonstrator is **irrational** if its planner returns policies with lower expected value than the optimal policy, for at least one $\theta \in \Theta$.

### 2.3 TYPES AND DEGREES OF IRRATIONALITY

There are many possible irrationalities that people exhibit (Chipman, 2014), far more than what we could study in one paper. To provide good coverage of this space, we start from the Bellman update, and systematically manipulate its terms and operators to produce a variety of different irrationalities that deviate from the optimal MDP policy in complementary ways. For instance, operating on the discount factor can model myopic behavior, while operating on the transition function can model optimism or the illusion of control. We parametrize each irrationality so that we can manipulate its "intensity" or deviation from rationality. Figure 2 summarizes our approach, which we detail below.

### 2.3.1 RATIONAL DEMONSTRATOR

In our setup, the **rational** demonstrator does value iteration using the Bellman update from Fig. 2. Our models change this update to produce different types of non-rational behavior.

### 2.3.2 MODIFYING THE MAX OPERATOR: BOLTZMANN

**Boltzmann**-rationality modifies the maximum over actions $\max_a$ with a Boltzmann operator with a parameter $\beta$:

$$V_{i+1}(s) = \text{Boltz}_a^\beta \sum_{s' \in S} P_{s,a}(s')\left(r_\theta(s, a, s') + \gamma V_i(s')\right),$$

where $\text{Boltz}^\beta(\mathbf{x}) = \sum_i x_i e^{\beta x_i} / \sum_i e^{\beta x_i}$ (Ziebart et al., 2010; Asadi and Littman, 2017). This is the most popular stochastic model used in reward inference(Ziebart et al., 2010; Asadi and Littman, 2017; Fisac et al., 2017). After computing the value function, the Boltzmann-rational planner $d_{\text{Boltz}}$ returns a policy where the probability of an action is proportional to the exponential of the $Q$-value of the action:

$$\pi(a|s) \propto e^{\beta Q_\theta(s, a)}.$$

The constant $\beta$ is called the *rationality constant*, because as $\beta \to \infty$, the human choices approach perfect rationality (optimality), whereas $\beta = 0$ produces uniformly random choices.

### 2.3.3 MODIFYING THE TRANSITION FUNCTION

Our next set of irrationalities manipulate the transition function away from reality.

**Illusion of Control.** People often overestimate their ability to control random events (Thompson, 1999). To model this, we consider demonstrators that use the Bellman update:

$$V_{i+1}(s) = \max_a \sum_{s' \in S} P_{s,a}^n(s')\left(r_\theta(s, a, s') + \gamma V_i(s')\right)$$

where $P_{s,a}^n(s') \propto (P_{s,a}(s'))^n$. As $n \to \infty$, the demonstrator acts as if it exists in a deterministic environment. As $n \to 0$, the demonstrator acts as if it had an equal chance of transitioning to every possible successor state.

**Optimism/Pessimism.** Many people systematically overestimate or underestimate their chance experiencing of positive over negative events (Sharot et al., 2007). We model this using demonstrators that modify the probability they get outcomes based on the value of those outcomes:

$$V_{i+1}(s) = \max_a \sum_{s' \in S} P_{s,a}^\omega(s')\left(r_\theta(s, a, s') + \gamma V_i(s')\right)$$

where $P_{s,a}^\omega(s') \propto P_{s,a}(s')e^{\omega\left(r_\theta(s,a,s') + \gamma V_i(s)\right)}$. $\omega$ controls how pessimistic or optimistic the demonstrator is. As $\omega \to +\infty$ (respectively, $\omega \to -\infty$), the demonstrator becomes increasingly certain that good (bad) transitions will happen. As $\omega \to 0$, the demonstrator approaches the rational demonstrator.

### 2.3.4 MODIFYING THE REWARD: PROSPECT BIAS

Next, we consider demonstrators that use the modified Bellman update:

$$V_{i+1}(s) = \max_a \sum_{s' \in S} P_{s,a}(s')\left(f(r_\theta(s, a, s')) + \gamma V_i(s')\right)$$

where $f : \mathbb{R} \to \mathbb{R}$ is some scalar function. This is equivalent to solving the MDP with reward $f \circ r_\theta$, and allows us to model human behavior such as loss aversion and scope insensitivity.

**Prospect Bias.** Kahneman and Tversky (2013) inspires us to consider a particular family of $f$s:

$$f_c(r) = \begin{cases} \log(1 + |r|) & r > 0 \\ 0 & r = 0 \\ -c\log(1 + |r|) & r < 0 \end{cases}$$

$c$ controls how loss averse the demonstrator is. As $c \to \infty$, the demonstrator primarily focuses on avoiding negative rewards. As $c \to 0$, the demonstrator focuses on maximizing positive rewards.

### 2.3.5 Modifying the sum between reward and future value: Extremal

**Extremal.** People seem to exhibit duration neglect, sometimes only caring about the maximum intensity of an experience (Do et al., 2008). We model this using Bellman update:

$$V_{i+1}(s) = \max_a \sum_{s' \in S} P_{s,a}(s') \max \left( r_\theta(s, a, s'), (1 - \alpha) r_\theta(s, a, s') + \alpha V_i(s') \right)$$

These demonstrators maximize the expected maximum reward along a trajectory, instead of the expected sum of rewards. As $\alpha \to 1$, the demonstrator maximizes the expected maximum reward received along the full trajectory. As $\alpha \to 0$, the demonstrator becomes greedy and maximizes the immediate reward.

### 2.3.6 Modifying the discounting

**Myopic Discount.** In practice, people are sometimes myopic, only considering near-term rewards. One way to model this is to decrease gamma in the Bellman update. At $\gamma = \gamma^*$, the discount rate specified by the environment, this is the rational demonstrator. As $\gamma \to 0$, the demonstrator becomes greedy and only acts to maximize immediate reward.

**Myopic Value Iteration.** As another way to model human myopia, we consider a demonstrator that performs only $h$ steps of Bellman updates. That is, this demonstrator cares equally about rewards for a horizon $h$, and discount to 0 reward after that. As $h \to \infty$, this demonstrator becomes rational. If $h = 1$, this demonstrator only cares about the immediate reward.

**Hyperbolic Discounting.** Human also exhibit hyperbolic discounting, with a high discount rate for the immediate future and a low discount rate for the far future (Grüne-Yanoff, 2015). Alexander and Brown (2010) formulate this as the following Bellman update:

$$V_{i+1}(s) = \max_a \sum_{s' \in S} P_{s,a}(s') \frac{r_\theta(s, a, s') + V_i(s')}{1 + k V_i(s') s}$$

$k$ modulates how much the demonstrator prefers rewards now versus the future. As $k \to 0$, this demonstrator becomes a rational demonstrator without discounting.

## 3 Exploring the effects of known biases on reward inference

Armed with our framework for characterizing irrationality, we test its implications for reward inference. We start by investigating the effects in random MDPs.

### 3.1 Experimental Design: Exact Inference in Random MDPs.

**Independent Variables.** We manipulate the type of the planner, and vary the degree parameters for each. We use different environments, sample different ground truth reward parameters, and test different trajectory lengths for the demonstrated behavior ($T = 3$, 15, and 30 state-actions pairs).

**Dependent Measures.** To separate the inference difficulty caused by suboptimal inference from the difficulty caused by demonstrator irrationality, we perform the exact Bayesian update on the trajectory $\xi$ (Ramachandran and Amir, 2007), which gives us the posterior on $\theta$ given $\xi$, $P(\theta|\xi) = \frac{P(\xi|\theta)P(\theta)}{\int_{\theta'} P(\xi|\theta')P(\theta')}$. Our primary metric is the expected **log loss** of this posterior:

$$\text{Log Loss}(\theta|\xi) = E_{\theta^*, \xi \sim d(\theta^*)} \left[ -\log P(\theta^*|\xi) \right].$$

A low log loss implies that we are assigning a high likelihood to the true $\theta^*$. Note that in this case, the log loss is equal to the entropy of the posterior $H(\theta|\xi) = H(\theta) - \mathbf{I}(\theta; \xi)$.

For each environment and irrationality type, we calculate the performance of reward inference on trajectories of a fixed length $T$. To sample a trajectory of length $T$ from a demonstrator, we fix $\theta^*$ and start state $s$. Then, we generate rollouts starting from state $s$ until $T$ state, action pairs have been sampled from $\pi = d(\theta^*)$. We repeat this procedure 10 times for each start state.

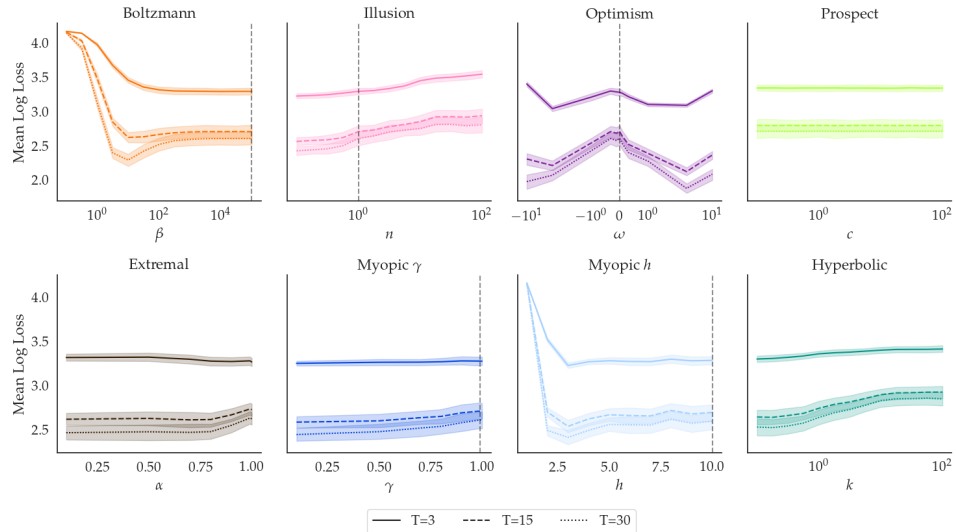

Figure 3: The log loss (lower = better) of the posterior as a function of the parameter we vary for each irrationality type, on the random MDP environments. For the irrationalities that interpolate to the rational planner, we denote the value that is closest to rational using a dashed vertical line. Every irrationality except Prospect Bias all have parameter settings that outperform the rational planner. The error bars show the standard error of the mean, calculated by 1000 bootstraps across environments.

**Simulation Environment.** We used MDPs with 10 states and 2 actions, where each $(s, a)$-pair has 2 random, nonzero transition-probabilities. We used $\gamma = 0.99$ and start trajectories in every state without reward. In these $\theta$ is a vector of length 3, where $\theta_i$ is the reward received from transitions out of state $i$ (and all other rewards are 0). We discretized each $\theta_i$ with 4 values, leading to $|\Theta| = 64$. We generated 20 such random MDPs.

## 3.2 RESULTS

Fig. 3 plots the log loss for each irrationality for random MDPs. The degree affects reward inference, with many settings naturally resulting in worse inference, especially at the extremes. However, what is surprising is that *every* type except Prospect Bias has at least one degree (parameter) setting that results in *better* inference with enough data: we see that most irrationality types can be more informative than rational behavior. The more data we have, the more drastic the difference (T=30 results in both better inference and larger difference relative to rational).

We put this to the test with a repeated-measures ANOVA with planner type as the independent variable, using the data from the best parameter setting from each type and T=30, and environment as a random effect. We find a significant main effect for planner type ($F(8, 806372) = 6102.93$, $p < 0.001$). A post-hoc analysis with Tukey HSD corrections confirmed that every irrationality except Prospect improved inference compared to the fully rational planner ($p < .001$). For $T = 30$, Optimism with $\omega = 3.16$ performed best, followed by Illusion of Control with $n = 0.1$ and Boltzmann with $\beta = 10$.

## 4 DOES THIS EFFECT GENERALIZE?

In the domain of random MDPs, we found that not only does irrationality affect reward inference, certain irrationalities can actually result in better inference. In this section, we probe the generality of this finding empirically, and explain it theoretically.

## 4.1 GRIDWORLD

Random MDPs lack structure, so we first test a toy environment that adds natural structure based on OpenAI Gym's 'Frozen-Lake-v0' (Brockman et al., 2016) (Fig. 5 in the appendix). Fig. 6 in the appendix shows the results in the gridworld: they are eerily (and reassuringly) similar to the random MDPs. This suggests that irrationalities can indeed help inference across differently structured MDPs. By inspecting the policies, we see that the irrational demonstrators were able to outperform the

Table 1: Autonomous Driving Results: Merging

| | Bayesian IRL | | CIOC | |
|---|---|---|---|---|
| Horizon | Log Loss | Information Gain | Cosine Similarity | Normalized Return |
| 3 | **0.690** | **0.696** | **0.999** | 0.939 |
| 5 | 0.824 | 0.562 | 0.940 | **0.981** |
| 7 | 1.383 | 0.004 | 0.350 | 0.856 |

rational demonstrator by disambiguating between $\theta$s that the rational demonstrator could not. To visualize this, we show examples of how the policy of several irrational demonstrators differ on the gridworld when the rational demonstrator's policies are identical in Fig. 7, Fig. 8, and Fig. 9 (included in the appendix).

## 4.2 AUTONOMOUS DRIVING

Even if known irrationality helps reward inference both empirically and theoretically in small MDPs, the question still remains whether this result will matter in practice. Next, we investigate the effect of demonstratsor irrationality on reward inference in a continuous state and action self-driving car domain (Sadigh et al., 2016). Switching to this real-world domain means we can no longer plan exactly, so we use model predictive control. It also means we can no longer run exact inference, so we approximate Bayesian IRL through samples, and also test continuous IOC (CIOC, Levine and Koltun (2012)) which recovers an approximate MLE. We measure the cosine similarity between the estimate and $\theta^*$, as well as the normalized reward of the trajectory when optimizing the recovered estimate. We use the merging task from Fig. 1 (more details in the appendix).

We report our results in Table 1. As the columns relating to Bayesian IRL show, decreasing the MPC planning horizon significantly decreases the log loss of the Bayesian IRL posterior. As with the gridworld results in section 4.1, the reason for this is that the shorter planning horizons exhibit more diverse behavior (as a function of the reward). Of the 4 reward settings we used, MPC with horizon 3 and 5 produced two different qualitative behaviors, whereas all rewards led to qualitatively similar behavior with horizon 7 (depicted in Fig. 1). When the weight on target speed was large enough, MPC with horizon 3 and 5 would produce a trajectory that overtakes the front car by going off the road. In all other cases, the demonstrator would merge between the two other cars.

We found similar results for CIOC: decreasing the MPC planning horizon increases the cosine similarity between the true and CIOC-recovered rewards. The return of optimizing the CIOC-recovered rewards is also higher when the demonstrator's planning horizon is 3 or 5 than 7.

Overall, yet again we find that irrationality (in this case specifically myopia) improves reward inference by producing more diverse behavior as a function of the reward.

## 4.3 THEORETICAL ANALYSIS

We now investigate this phenomenon theoretically. We show that not only can rational behavior be arbitrarily less informative than irrational behavior, but also that this applies to Boltzmann-rationality.

**Informativeness as mutual information.** The mutual information $\mathbf{I}(\theta; d(\theta)) = H(\theta) - H(\theta|d(\theta))$ between the policy and the reward parameters allows us to quantify the informativeness of a planner. As the conditional entropy $H(\theta|d(\theta)) = E\left[-\log p(\theta|d(\theta))\right]$ is equal to the log loss of the posterior under the true model, the mutual information upper bounds how much better (in terms of log loss) *any* inference procedure can do, relative to the prior. A known insight is that planners that optimize for maximizing the discounted sum of rewards *are not the same* as those that optimize for being informative (Dragan et al., 2013; Hadfield-Menell et al., 2016): $\arg\max_d E_\theta[V_\theta^{d(\theta)}] \neq \arg\max_d \mathbf{I}(\theta; d(\theta))$. While this insight has been tested empirically, we begin with a theoretical lens for understanding it.

**Irrationalities exist that are arbitrarily better for inference than rationality.** We first consider **deterministic planners**: planners that return deterministic policies. We show that there are cases where the rational behavior is not informative at all, whereas some (irrational) deterministic planner achieves the theoretical upper bound on informativeness. (Proof in appendix.)

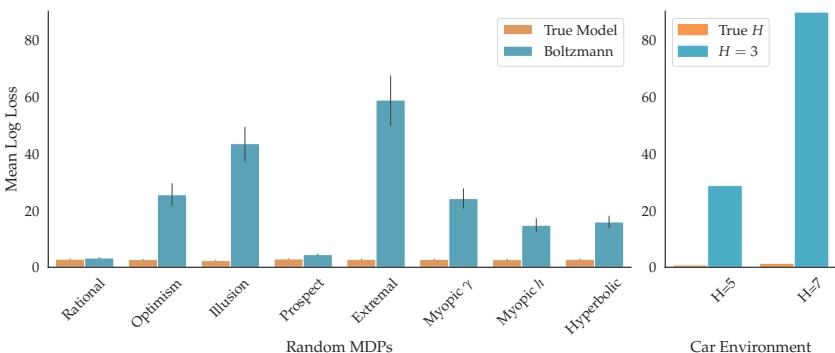

Figure 4: A comparison of reward inference using a correct model of the irrationality type, versus always using a Boltzmann-rational model ($\beta = 10$), on the random MDPs (left) and the car environment (right). The impairment due to model misspecification greatly outweighs the variation in inference performance caused by various irrationalities. The error bars show the standard error of the mean, calculated by the bootstrap across environments.

**Proposition 1.** *There exists a family of URMDPs with state spaces of any size, such there exists a deterministic planner $d_{Irrational}$ satisfying $\mathbf{I}(\theta, d_{Irrational}(\theta)) = \log|\Theta|$ and $\mathbf{I}(\theta, d_{Rational}(\theta)) = 0$.*

**Boltzmann-rationality is (arbitrarily) more informative than full rationality.** Of course, the upper bound is attained by *some* irrational planner. It demonstrates that an demonstrator can perform better than rational when specifically optimizing for informativeness. But this is an artificial, contrived kind of irrationality. In fact, prior work that maximized informativeness did so by solving a *more difficult* problem than rationality (Dragan et al., 2013; Hadfield-Menell et al., 2016). Here, we provide evidence that Boltzmann-rationality, a standard model of stochastic choice (discussed in section 2.3), outperforms full rationality for reward inference. (Proof in appendix.)

**Proposition 2.** *There exists a family of one-state two-action MDPs, with arbitrarily large $|\Theta|$ such that $\mathbf{I}(\theta, d_{Boltz}(\theta)) = \log|\Theta|$ and $\mathbf{I}(\theta, d_{Rational}(\theta)) = 0$.*

## 5 EFFECTS OF MISSPECIFICATION ON REWARD INFERENCE

We see that irrationalities sometimes hinder, but sometimes help reward inference. So far, the learner had access to the type (and degree of irrationality) during inference. Next, we ask how important it is to know this. Can we not bother with irrationality, make a default assumption, and run inference?

**Assuming noisy rationality can lead to very poor inference.** Fig. 4 suggests that the answer is no. We start by comparing inference with the true model on random MDPs versus with assuming the standard Boltzmann model as a default. The results are quite striking: not knowing the correct irrationality harms inference tremendously.[1] We then confirm that misspecification greatly harms inference in the autonomous driving environment. This emphasizes the importance of understanding irrationality when doing reward inference going forward.

**Approximate models of irrationality might be enough.** This finding is rather daunting, as perfect models of irrationality are very challenging to develop. But do they need to be perfect? Our final analysis suggests that the answer is no as well. In Fig. 11 of the appendix, we report the log loss of inference with the correct type, but under misspecification of the parameter. Encouragingly, we find that in many cases, merely getting the type of the demonstrator irrationality correct is actually sufficient to lead to much better inference than assuming Boltzmann rationality. Further, we also find evidence that the learner does not need to get the type exactly right either: as shown in Fig. 12, if the learner accounts for the demonstrator's myopia, but gets the type of the myopia wrong, this still leads to significantly better inference than assuming Boltzmann rationality.

---

[1](Shah et al., 2019) proposed a way to model irrationality and analyzed its benefit over assuming Boltzmann; the benefit was very limited, which they attributed due to their deep learning model's brittleness compared to exact planning. Here, we compare to perfect modeling to analyze the headroom that modeling has.

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

# Appendices

## A    DETAILS FOR GRIDWORLD

In this section, we investigate the effects of irrationality on inference in an MDP based on OpenAI Gym's 'Frozen-Lake-v0' (Brockman et al., 2016). This a small 5x5 gridworld (Fig. 5), consisting of three types of cells: ice, holes, and rewards. The demonstrator can start in any ice cell. At each ice cell, the demonstrator can move in one of the four cardinal directions. With probability 0.8, they will go in that direction. With probability 0.2, they will instead go in one of the two adjacent directions. Holes and rewards are terminal states, and return the demonstrator back to their start state. They receive a penalty of $-10$ for falling into a hole and $\theta_i \in [0, 4]$ ($|\Theta| = 25$) for entering into the $i$th reward cell.

Fig. 6 shows the analogue of the results in section 3.2 for this gridworld: they are reassuringly similar. We use the same experimental design as the random MDPs in section 3.1.

To visualize why inference quality is improved, we show examples of how the policy of several irrational demonstrators differ on the gridworld when the rational demonstrator's policies are identical in Fig. 7, Fig. 8, and Fig. 9.

Finally, Fig. 10 example of why using the wrong model for reward inference leads to bad inference. In it, the reward inference algorithm assumes that the demonstrator in Boltzmann when it is actually Myopic. The Boltzmann rational agent would take this trajectory only if the reward at the bottom was not much less than the reward at the top. The myopic agent with $n \leq 4$, however, only "sees" the reward at the bottom. Consequently, inferring the preferences of the myopic agent as if it were Boltzmann leads to poor performance in this case.

## B    DETAILS FOR AUTONOMOUS DRIVING ENVIRONMENT

Even if known irrationality helps reward inference both empirically and theoretically in small MDPs, the question still remains whether this result will matter in practice. As a result, we investigate the effect of demonstrator irrationality on reward inference in the self-driving car domain (Sadigh et al., 2016).

**Simulation environment.** As in previous work in the car domain (Sadigh et al., 2016; 2017), we model the dynamics of cars using a point-mass model. The state of each car is a 4-dimensional vector $s = [x \ y \ h \ v]$, where $x, y$ are the coordinates of the car, $h$ is the heading, and $v$ is the speed. The control input for the car is a two dimensional vector $a = [u_1 \ u_2]$, where $u_1$ is the steering input and

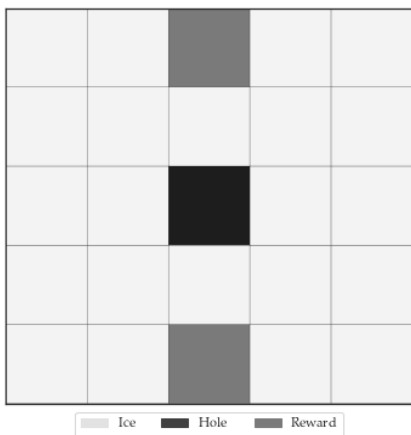

Figure 5: The gridworld used in section 4.1.

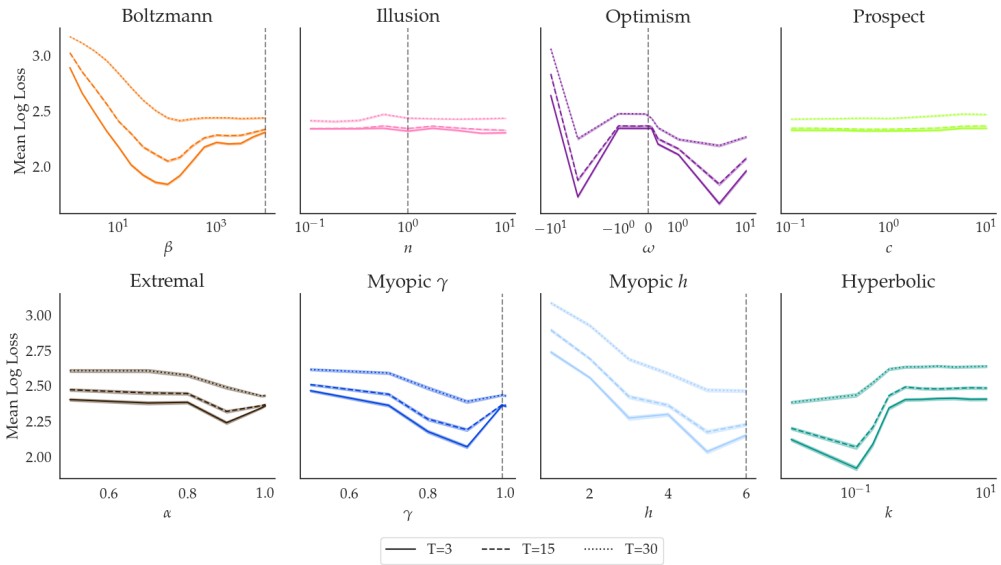

Figure 6: The analog of Fig. 3 for the gridworld. Error bars are the standard error of the mean. The findings are surprisingly similar as with the random MDPs. Note the more limited x-axis ranges we used in this experiment.

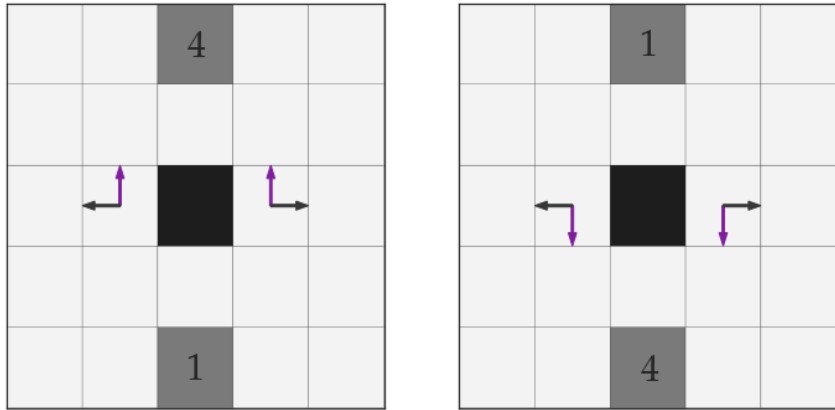

Figure 7: Optimism bias ($\omega = 3.16$) produces different actions for $\theta^* = (4, 1)$ vs. $\theta^* = (1, 4)$ in the states shown: the rational policy is to go away from the hole regardless of $\theta$, but an optimistic demonstrator takes the chance and goes for the larger reward – up in the first case, down in the second.

$u_2$ is the acceleration. We also include a friction coefficient $\alpha$. The dynamics model of the vehicle is:

$$[\dot{x}\ \dot{y}\ \dot{h}\ \dot{v}] = [v \cdot \cos h \quad v \cdot \sin h \quad v \cdot u_1 \quad u_2 - \alpha \cdot v].$$

For ease of simulation, we discretized the simulation along the time dimension using the following dynamics:

$$\Delta s_t = [\Delta x_t\ \Delta y_t\ \Delta h_t\ \Delta v_t] = [\bar{v}_t \cdot \cos h_t \cdot dt \quad \bar{v}_t \cdot \sin h_t \cdot dt \quad \bar{v}_t \cdot u_1 \cdot dt \quad (u_2 - \alpha \cdot v_t) \cdot dt],$$

where $\Delta s_t = s_{t+1} - s_t$ and $\bar{v}_t = v_t + 0.5 \cdot u_2 \cdot dt$. In our experiments, we used $dt = 0.1s$.

Reward functions in this environment are assumed to be a linear combination of features:

$$r_\theta(s, a) = \theta^\top f(s, a), \ \ f : S \times A \to \mathbb{R}^n$$

Due to environment complexity, we can't solve for optimal trajectories in our environment directly. Instead, we suppose that the planner of is performing Model Predictive Control (MPC) at every

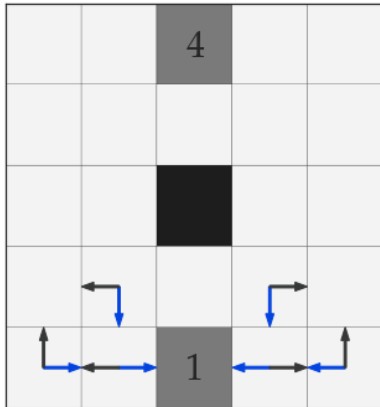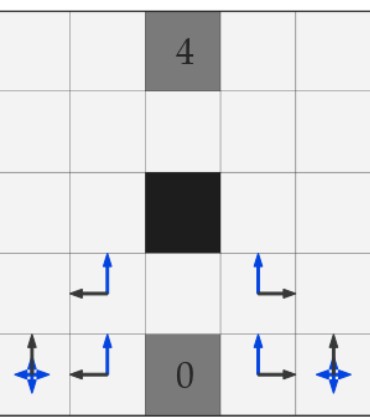

Figure 8: Myopic value iteration ($h = 5$) produces different policies for $\theta^* = (4, 1)$ vs. $\theta^* = (4, 0)$: while the rational expert always detours around the hole and attempts to reach the larger reward, myopia causes the myopic expert to go for the smaller source of reward when it is non-zero.

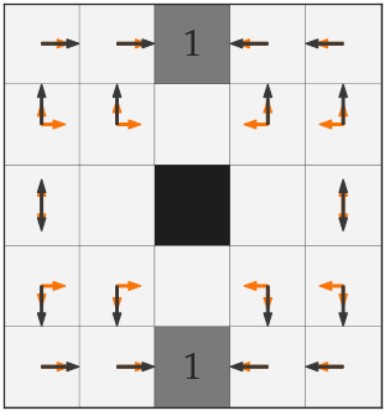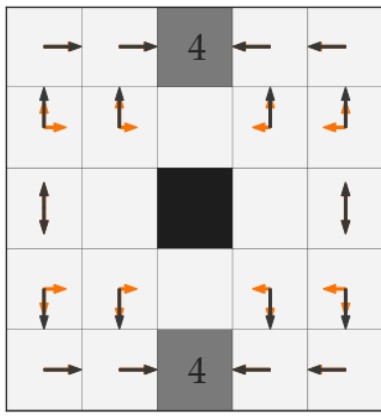

Figure 9: Boltzmann ($\beta = 100$) produces different policies for $\theta^* = (1, 1)$ vs. $\theta^* = (4, 4)$: when $||\theta||$ is larger, the policy becomes closer to that of the rational demonstrator, as the differences in $Q$-values becomes larger.

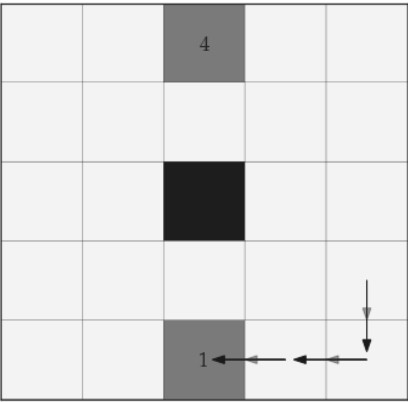

Figure 10: An example of why assuming Boltzmann is bad for a myopic agent - the Boltzmann rational agent would take this trajectory only if the reward at the bottom was not much less than the reward at the top. The myopic agent with $n \leq 4$, however, only "sees" the reward at the bottom. Consequently, inferring the preferences of the myopic agent as if it were Boltzmann leads to poor performance in this case.

iteration – that is, it will plan a finite horizon sequence of actions to maximize its reward, execute the first action in the sequence, then replan. As a basic model of irrationality, we consider shorter planning horizons. We assume that the reward inference procedure knows the planninng horizon exactly.

**Bayesian IRL in the car environment** First, we consider the analogue of the results in section 3.2 in the car environment: what is the log-loss of the posterior on $\theta$, given different planning horizons? Since the demonstrations used in this work are generated via trajectory optimization (and not an inherently stochastic process), using Bayesian IRL requires us to specify a "fake" observation model $P(\xi|\theta)$.

We use the following distribution, normal in feature space, for $P(\xi|\theta)$:

$$P(\xi|\theta) \propto e^{(E[f(\xi)] - E[f(\xi_\theta^*)])^T \Sigma_{\xi_\theta^*}^{-1} (E[f(\xi)] - E[f(\xi_\theta^*)])},$$

where $E[f(\xi_\theta^*)]$ is the expectation of the features of the optimal trajectory $\xi_\theta^*$ for $r_\theta$.

**Maximum likelihood IRL in cars** In practice, full Bayesian inference is completely intractable for complicated domains such as cars. Instead, the state of the art for reward inference are approximate, maximum-likelihood estimate (MLE) based methods.To study the effects of irrationality on MLE-based inference methods, we also perform Levine and Koltun's Continuous IOC with Locally Optimal Examples (CIOC) (Levine and Koltun, 2012), which uses a Boltzmann model of the demonstrator with a second order Laplace approximation of the normalizing constant. Since our demonstrators are actually locally optimal, we rectify the bias induced by the Boltzmann model by using a large $\beta = 10^4$. To rectify the issue of local optimization in CIOC, we initialize the optimization procedure with the true reward weights.

As CIOC returns an (approximate) MLE estimate for the reward parameter $\theta$, we cannot use the log-loss metric. Instead, we evaluate the reward functions by the **cosine similarity** of $\theta^*$ and $\hat{\theta}$, as well as the **normalized return** of the trajectory when optimizing the recovered $\hat{\theta}$. (We normalize the returns so that the optimal trajectory has return $1$ and the trajectory that goes forward at constant speed $0$.)

**Driving Scenario: Merging** Our experiments were performed in a simple merging environment (Fig. 1). In it, the demonstrator wants to merge into the right lane while trying to maintain its $1.2$ forward speed. In addition to the demonstrator car, the right lane contains two constant velocity cars, traveling at $0.8$ speed. The features of this environment are composed of a squared penalty for deviating from $1.2$ forward speed, features for the squared distances to the medians of each of the lanes, a feature for the minimum squared distance to any of the medians of the lanes, and a smooth collision feature.

For the Bayesian IRL scenario, we considered four different reward functions on this domain, consisting of varying the weight on the penalty for deviating from the target speed. All the other weights are unchanged. In particular, we considered $\theta_{\text{speed}} \in \{0.5, 1, 2, 4\}$. In the CIOC scenario, we used $\theta_{\text{speed}} = 1$. We also consider 3 different planning horizons: $h = 3$, $h = 5$, and $h = 7$. This means we had 12 conditions for Bayesian IRL and 3 for CIOC.

## C   PROOFS FOR SECTION 4.3

First, note that for deterministic planners, there that there are $|A|^{|S|}$ such policies, and thus we have $\mathbf{I}(\theta; d(\theta)) \leq H(d(\theta)) \leq |S| \log |A|$ (as the entropy of a discrete random variable $X$ is bounded above by the logarithm of the size of its support $|X|$). Similarly, we also have that $\mathbf{I}(\theta; d(\theta)) \leq H(\theta) \leq \log |\Theta|$. For deterministic planners, we thus have

$$\mathbf{I}(\theta; d(\theta)) \leq \min(\log |\Theta|, |S| \log |A|),$$

We will now prove a stronger version of proposition 3.

**Proposition 3.** *There exists a family of URMDPs with state and action spaces of any size, such that the rational planner provides no information, and there exists a deterministic planner that provides $min(\log |\Theta|, |S| \log |A|)$ bits worth of information.*

*Proof.* Consider a set of environments where $\Theta = \{1, 2, ..., |A|^{|S|}\}$, the prior $p$ is uniform over $\Theta$ (and thus $H(\theta) = \log |\Theta|$), and where

$$r_\theta(s, a, s') = \begin{cases} \theta & a = a^* \\ 0 & a \neq a^* \end{cases}$$

for some $a^*$. Then the unique optimal policy has $\pi(s) = a^*$ for every $\theta$ (as $\theta > 0$). This implies that

$$\mathbf{I}(\theta, d_{\text{Rational}}(\theta)) \leq H(d_{\text{Rational}}(\theta)) = 0.$$

However, the planner $d' : \theta \mapsto \pi^{(\theta)}$, with some fixed some ordering $\{\pi^{(i)}\}_{i \in \{1,2,...,|A|^{|S|}\}}$ of the possible policies, satisfies $H(\theta | d'(\theta)) = 0$ and thus $\mathbf{I}(\theta; d'(\theta)) = H(\theta) = \log |\Theta| = |S| \log |A|$. $\square$

By choosing $A$ such that $|A| \geq |\theta|^{1/|S|}$, we get the Proposition 3 in the text.

**Proposition 4.** *There exists a family of one-state two-action MDPs, with arbitrarily large $|\Theta|$ such that $\mathbf{I}(\theta, d_{Boltz}(\theta)) = \log |\Theta|$ and $\mathbf{I}(\theta, d_{Rational}(\theta)) = 0$.*

*Proof.* Suppose first action $a_1$ has reward $\theta$, where $\theta \in \{1, 2, ..., -|\Theta|\}$, and the second action $a_2$ has reward 0. Then the rational planner $d_{\text{Rational}}$ will always return a policy that always takes the first action, while $d_{\text{Boltz}}(\theta)$ is the policy

$$\pi_\theta(a_1|s_1) = \frac{e^{\beta(\theta + V(s_1))}}{e^{\beta(\theta + V(s_1))} + e^{\beta V(s_1)}} = \frac{1}{1 + e^{-\beta\theta}}$$

$$\pi_\theta(a_2|s_1) = \frac{1}{1 + e^{\beta\theta}}$$

This mapping is injective, and so $H(\theta | d_{\text{Boltz}}(\theta)) = 0$ and thus $\mathbf{I}(\theta; d_{\text{Boltz}}(\theta)) = H(\theta) = \log |\Theta|$. $\square$

## D  MORE THEORY

**Why recover the reward parameter?**  Even if irrationalities can help with inference, a natural question why we wish to infer the reward. If the actual goal is acting optimally, a rational demonstrator would be great because we can just imitate them. In addition to the fact that real people are not perfectly rational, the reason we focus on inference is that imitation is not enough in some cases, e.g. if we need to behave in new environments (Taylor and Stone, 2009; Devin et al., 2017; Jing et al., 2019), or transfer the policy to a robot with different dynamics (Cully et al., 2015; Fu et al., 2018; Reddy et al., 2018).

In fact, there are cases where being robust to changes in dynamics requires identifying the reward parameters. Indeed, there exist such cases where it is impossible to identify the parameters from rational demonstrators, but possible from Boltzmann demonstrators:

**Proposition 5.** *There exists an URMDP $M$ and a set of new transition probabilities $\{\{P_{s,a}^{(i)}\}\}_{i \in \{1,2,...,|\Theta|\}}$ such that $\log |\Theta|$ bits of information are needed to compute the optimal policy under all the transition probabilities; and where $\mathbf{I}(\theta, d_{Rational}(\theta)) = 0$, whereas $\mathbf{I}(\theta, d_{Boltz}(\theta)) = \log |\Theta|$.*

*Proof.* We construct a 2 state, 2 action URMDP. Label the two states $s_1$ and $s_2$. Let $\Theta = \{1, 2, ..., |\Theta|\}$, the prior be uniform over $\Theta$, and let $r(s, a, s_1) = \theta$ for all $s, a$ and $r(s, a_1, s_2) = 0$, $r(s, a_2, s_2) = -1$ for all $s$.
Under the original transition probabilities $\{P_{s,a}\}$, $a_1$ leads deterministically to $s_1$ while $a_2$ deterministically leads to $s_2$. Then, the unique optimal policy is $\pi(s) = a^*$ for every $\theta$ (as $\theta > 0$). This implies that $\mathbf{I}(\theta, d_{\text{Rational}}(\theta)) = 0$. On the other hand, $d_{\text{Boltz}}$ is an injective map (as in Proposition 2), and thus $\mathbf{I}(\theta; d_{\text{Boltz}}(\theta)) = \log |\Theta|$.
Now, we construct $\{P_{s,a}^{(i)}\}$ such that $a_1$ is optimal for $\theta \leq i - 1$, and $a_2$ is optimal for $\theta \geq i$. First, let $a_1$ deterministically lead to $a_2$, thus, leading to reward 0. Next, let $a_2$ lead to $s_1$ with probability $\frac{2}{2i+1}$. Therefore:

$$E_{s' \sim P_{s,a_2}^{(i)}}[r_\theta(s, a_2, s')] = \frac{2}{2i+1}\theta - \frac{2i-1}{2i+1}$$

Then for $\theta \geq i$, we have $E_{s' \sim P_{s,a_2}^{(i)}}[r_\theta(s, a_2, s')] \geq \frac{1}{2i+1} > 0$ and for $\theta \leq i - 1$, we have $E_{s' \sim P_{s,a_2}^{(i)}}[r_\theta(s, a_2, s')] \leq -\frac{1}{2i+1} < 0$.

By construction, in order to compute the optimal policy for every new transition probabilities $\{P_{s,a}^{(i)}\}$, we must know the value of $\theta$, and thus need $\log |\Theta|$ bits of information. $\qquad\square$

## E    ADDITIONAL MISSPECIFICATION RESULTS

**How well do we have to know the planners' parameters to outperform Boltzmann-rationality?**

A natural question to ask after the results of section 5 is how exactly we need to know the bias before we can outperform the Boltzmann-rational assumption. To explore this, we compare inference with the true planner model, but with (possibly) the wrong parameter in Fig. 11. In order to deal with the fact that deterministic planners can assign probability 0 to a trajectory, we smoothed the policies by applying a softmax to the planners' Q-values with $\beta = 10.0$. Surprisingly, we find that for many demonstrator irrationalities, a wide variety of assumed parameter settings still allowed the learner to outperform inference under the Boltzmann-rational assumption. In fact, for several irrationalities, the learner is always better off with the correct type, regardless of what planner parameter they assume. This suggests that we don't need to model human irrationalities perfectly to improve on existing reward inference.

**How well do we have to know the rationality type to outperform Boltzmann-rationality?**

Another natural question is whether we need to know the exact type of the planner. For example, while many pieces of evidence point toward people having some form of myopia in decision making, the exact form of the myopia may not be clear. To investigate this quesiton, we compare the learner's performance when the demonstrator assumes the *wrong* form of Myopia in Fig. 12. As before, we smoothed the policies by applying a softmax to the planners' Q-values with $\beta = 10.0$. We find that even though the learner gets the type of the irrationality wrong, the learner still does better than assuming Boltzmann-rationality. Again, this suggests that we don't need to model human irrationalities perfectly to improve on existing reward inference.

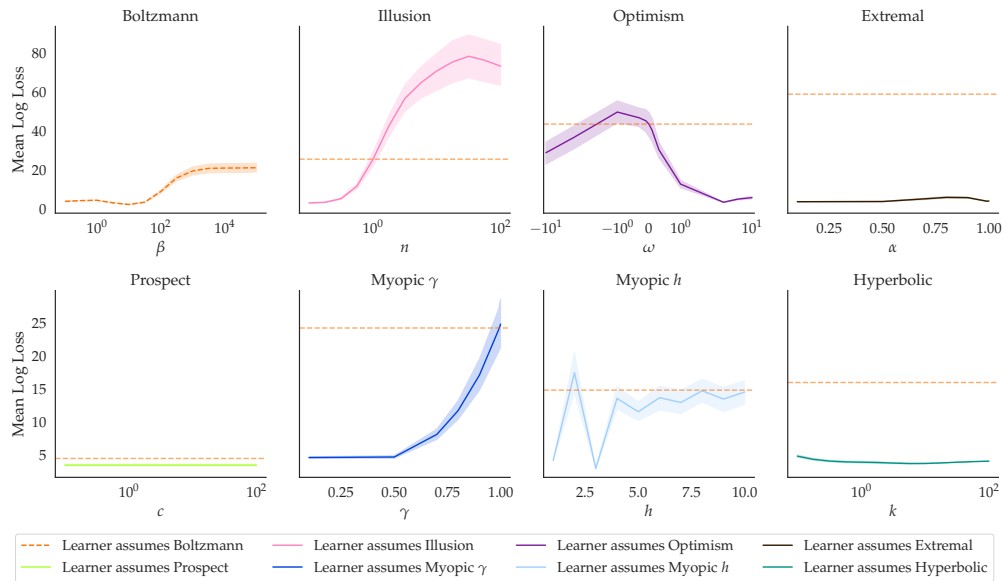

Figure 11: The log loss (lower = better) of various models under parameter misspecification. Each x-axis shows the parameter that the learner assumes. The orange line represents the performance when the learner makes the faulty assumption that the demonstrator is Boltzmann-rational. In many cases, the learners perform better than by assuming Boltzmann-rational just by getting the type of the planner correct, even if they don't get the exact parameter correct. The error bars show the standard error of the mean, calculated by the bootstrap across environments.

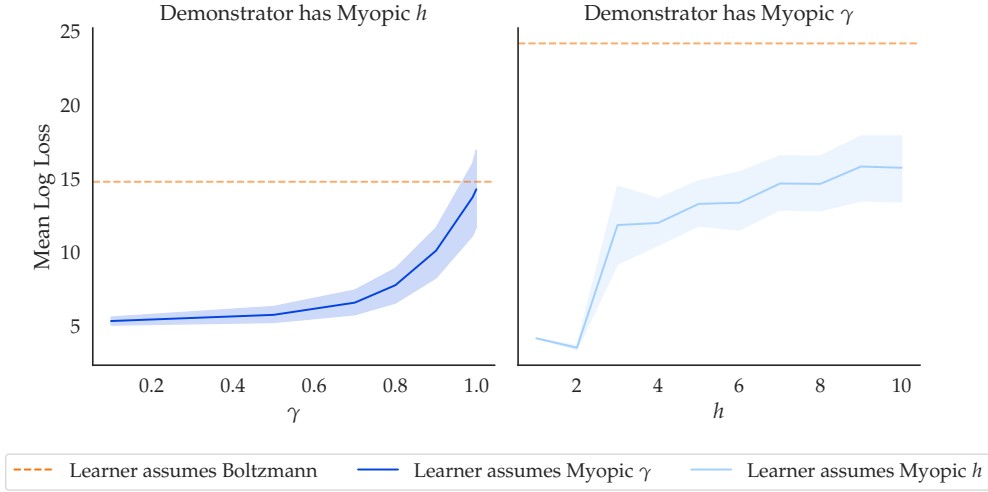

Figure 12: The log loss (lower = better) of two myopic demonstrators under type misspecification. On the left, the demonstrator performs myopic value iteration (Myopic $h$), but the learner assumes the demonstrator has a myopic discount rate $\gamma$ (Myopic $\gamma$). On the right, the demonstrator has a myopic discount rate $\gamma$ but the learner assumes myopic value iteration. However, in both cases, this leads to better inference than assuming Boltzmann-rationality. The error bars show the standard error of the mean, calculated by the bootstrap across environments.

