# OpenReview forum: "The impacts of known and unknown demonstrator irrationality on reward inference"
_ICLR.cc/2021/Conference — Reject_

### Official Review · AnonReviewer2 · 2020-10-27
**Interesting question and systematic manuscript outline, but concerns regarding applicability, triviality and absence of high-dimensional experiments**

**Rating:** 5
**Confidence:** 3

**Review:**

Summary

This paper investigates the effect different irrationality types have on reward inference. The irrationality types are modelled in the context of an Uncertain-Reward Markov Decision Process (URMDP) that is similar to an ordinary MDP but with a prior distribution over reward functions. Different irrationality types are expressed through different modifications of Bellman's optimality principle. In simple environment settings, given trajectories of an irrational agent with an optimal irrational policy (under the respective modified Bellman principle), Bayesian inference is feasible in order to identify a posterior over reward functions given the data (i.e. agent trajectories).

The authors provide a metric to evaluate the quality of the inference procedure. This metric is the expected log loss of the posterior evaluated at the optimal reward function (there is a further outer expectation over tasks with different reward functions). Based on this metric, the authors conclude that the quality of reward inference is higher for irrational agents than for fully rational agents, as evaluated in some grid world settings.

Quality and Details

I am on the verge when assessing the quality of this work. The manuscript tries to answer an interesting question and the outline and structure follow a systematic approach. However, I have 3 main concerns:

1.) While there are a bunch of irrationality models presented in the context of an URMDP, it is unclear how realistic these models are for actual reward inference in practical problems.

2.) I do appreciate both the empirical and theoretical analysis. However, I feel that the results may be trivial---especially in settings where an irrationality model can represent infinitely many decision-makers, one of which being a perfectly rational agent. I would intuitively argue that it is then quite "likely" that reward inference is easier for some of the infinitely many irrational agents as opposed to the one perfectly rational agent (recovered with a specific irrationality parameter value).

Similarly holds for the theoretical results. Proposition 1 is for example trivially true for a URMDP where the perfectly rational agent is the same for all reward functions, and where the optimal irrational agent assigns a different policy to each reward function. Proposition 2 seems to be a special case of the former for one-state-two-action settings and entropy-regularized irrationality. I simply don't know what to take away from these results...

3.) The experimental setting only considers low-dimensional grid worlds (as a recommendation, I would put more emphasis on the Frozen Lake setting rather than the simpler setting). Only autonomous driving is presented as a high-dimensional task---but in this context, irrationality is only considered in terms of the agent's planning horizon but not in terms of any of the irrationality types presented earlier for lower-dimensional settings.

Clarity

The paper is clearly written and easy to follow.

Originality and Significance

The question, the paper asks, is original but the significance is limited. This is because it is not clear how reasonable the proposed irrationality models are, and because high-dimensional experiments with the presented irrationality models are missing.

Pros

An exhaustive list of irrationality models is tested.

Cons

It is questionable how practically relevant the irrationality models, studied by this work, are. Results may be trivial and high-dimensional experiments are largely missing.


Minor

Some more explanation in some places would help. For example, when mentioning Bayes' rule in low-dimensional settings, it could be explained more clearly how the likelihood looks like, and how the specific irrationality model affects the likelihood (through the optimal irrational policy I assume?).

I would have also appreciated a bit more details about the lowest-dimensional setting: how exactly the reward parameter theta is chosen is a bit difficult to understand when reading the caption of Figure 3.

---

> ### Author Response · Authors · 2020-11-24
> **Response to Reviewer 2**
>
> Thanks for your detailed review! We’re happy that you found our paper well written and easy to follow, agree that an exhaustive list of irrationality models has been tested, and also that the problem we are trying to answer is interesting.
>
> We agree that our theoretical results are not necessarily surprising or technically difficult. Therefore, we place little emphasis on them, and instead use them to explain the more surprising (to us) empirical results. That being said:
> >  I would intuitively argue that it is then quite "likely" that reward inference is easier for some of the infinitely many irrational agents as opposed to the one perfectly rational agent...
>
> Indeed, while prop 1 makes this claim, prop 2 makes a stronger claim - that a particular, commonly used irrational planner model makes reward inference easier. And our empirical results show that *many* irrational planners make reward inference easier.
>
> In fact, under many reasonable parameterizations of irrational planners, ‘most’ irrational planners will map fewer reward parameters to the same policies, therefore producing more informative behavior. Intuitively, we can think of this as optimal behavior being constrained in ways that irrational behavior is not. We will consider formalizing and including this result in the paper.
>
> > It is questionable how practically relevant the irrationality models, studied by this work, are. Results may be trivial and high-dimensional experiments are largely missing.
>
> As with the other reviewers, this reviewer raises an important question of practicality of this finding: since we usually don’t know the human’s bias, and the inference procedures studied in this work don’t scale to more realistic, high dimensional problems. This is a valid point, but even if we can’t infer the bias and reward of humans today:
> * The finding itself has scientific value intrinsically, as it’s pointing out a surprising fact about biased behavior. Note that our paper is not a typical “here’s a new method” ML paper, it’s merely an analysis, one that arguably produced a surprising finding about a well-studied topic and therefore has value in itself.
> * The finding encourages taking biases seriously: not just because we make the wrong inference if we don’t, but because there is something to be gained by modeling them. It serves as an argument for further research in understanding and modeling enough about biases.  As other reviewers have pointed out, cognitive science, and behavioral economics have identified many biases. So while the approach in this work may not immediately scale, we believe the results can still inform research in this area.
> * The finding also points to future research directions where robots attempt to influence their demonstrators to exhibit a specific bias (like myopia) in order to make them more informative. While it’s really hard for people to be pedagogical, it might be much easier for them to act myopically. Of course, how to do this well and whether it works out is an open question, but our paper uncovers that as a potentially new avenue for improving learning.
>
>
> > The experimental setting only considers low-dimensional grid worlds (as a recommendation, I would put more emphasis on the Frozen Lake setting rather than the simpler setting). Only autonomous driving is presented as a high-dimensional task---but in this context, irrationality is only considered in terms of the agent's planning horizon but not in terms of any of the irrationality types presented earlier for lower-dimensional settings.
>
> Thanks for this suggestion. We can certainly place more emphasis on the Frozen Lake setting, though we will add that there are advantages to using random MDPs as well - for example, it suggests the results are more general than one or two particular gridworlds. We also considered analyzing more irrationalities in the autonomous driving task, but as the state is continuous and the planning is only local and approximate, the other irrationalities do not translate over as naturally as myopia.

---

### Official Review · AnonReviewer1 · 2020-10-27
**Good presentation of the formalism for a diverse set of irrationalities; experiments are lacking**

**Rating:** 5
**Confidence:** 3

**Review:**

This work studies the effect of modeling systematic irrationality of the demonstrator for reward learning problems. By manipulating different factors of Bellman update, the authors simulate different irrational behavior in demonstrations. Experiments in gridworld and a 2D driving domain demonstrate that modeling irrationality helps reward inference. The authors also demonstrate that knowing the general/approximate type of irrationality, instead of knowing the exact irrationality model, might be enough to improve reward inference.

Pros:
The problem setting is important. It is significant to model human rationality and irrationality for learning from the data generated by human demonstrators; insights from studying human behavior helps construct better learning algorithms.
This work formalizes a diverse set of irrationality, as observed in human behavior, within the MDP formalism for systematic analysis of the effect of perfect modeling
The presented formalism for irrationality centers around Bellman update; such a unified presentation of the formalism for irrationality is novel, prior works only presented a few types of irrationality separately as different modifications to the value-iteration algorithm

Cons:
It is not surprising that knowing the generative bias in data helps inference; assuming access to the underlying irrationality type is a strong assumption. The more important task is to understand what types of irrationality exist in natural human data and how to infer them.
This work only demonstrates that (in contrast to the findings of prior work of Shah et al.) knowing the true irrationality model outperforms assuming Boltzmann-rationality by a lot, however, it does not experimentally show the effect of assuming the wrong irrationality type on learning (besides Botlzmann-rational); if inferring the type of irrationality is efficient and approximately, then it is desirable to do so whenever possible; on the other hand, if assuming the wrong type of irrationality performs worse than assuming Boltzmann-rational, then it is important to know the risk.
The authors also acknowledge that it is hard to infer irrationality directly from human data and real human data may contain multiple types of irrationality; it is important to discuss how this influences the contribution of this paper along with results from the above point demonstrating the effect of assuming wrong or incomplete irrationality when multiple irrationalities exist.
In the abstract, it is indicated that the findings of “myopic behavior being more informative” allow us to ask human demonstrators to be myopic when they demonstrate; this deviates from the motivation to correctly model human’s irrationality in the data generation process but rather circumvent the problem of inferring human’s irrationality by conditioning the demonstrator to be “myopic”. Such claims should be avoided when there are no experimental results supporting their validity.

---

> ### Author Response · Authors · 2020-11-24
> **Response to Reviewer 1**
>
> We’d like to thank you for your review of our work. We are pleased that you agree with us that the problem setting is important, and that our formalism allows for a systematic analysis of the effects of irrationality on reward inference.
>
> The reviewer says:
> > It is not surprising that knowing the generative bias in data helps inference; assuming access to the underlying irrationality type is a strong assumption. … This work only demonstrates that (in contrast to the findings of prior work of Shah et al.) knowing the true irrationality model outperforms assuming Boltzmann-rationality by a lot ...
>
> We agree! Knowing the generative bias unsurprisingly helps with reward inference. What is more surprising (to us), though, is that biases can actually be helpful for inference. That is, a biased demonstrator can be more informative than a rational demonstrator, if the bias is known. (See our top-level comment, “Biases can be more informative - instead of merely important to model”.)
>
> > … however, it does not experimentally show the effect of assuming the wrong irrationality type on learning (besides Botlzmann-rational); if inferring the type of irrationality is efficient and approximately, then it is desirable to do so whenever possible; on the other hand, if assuming the wrong type of irrationality performs worse than assuming Boltzmann-rational, then it is important to know the risk.
>
> We agree that it is important to know the risks. We do investigate the effects of assuming incorrect irrationality parameters in section 5, and also investigate the effects of assuming the incorrect type of myopia.
>
> > In the abstract, it is indicated that the findings of “myopic behavior being more informative” allow us to ask human demonstrators to be myopic when they demonstrate; this deviates from the motivation to correctly model human’s irrationality in the data generation process but rather circumvent the problem of inferring human’s irrationality by conditioning the demonstrator to be “myopic”. Such claims should be avoided when there are no experimental results supporting their validity.
>
>
> We will soften our language, since we don’t want to claim that this will necessarily work. However we do want to point out the opportunity. While part of our work consists of investigating the costs of assuming the wrong irrationality type or parameter, another part of our work consists of investigating the effect of biases themselves. That is, we compare the situation where the human is biased in a particular way (and this bias is known) to the situation where the human is not biased in that way. Our results in that domain indicate that biased demonstrators can be more informative than an unbiased demonstrator, if the bias is known. (Again, see our top level comment, “Biases can be more informative - instead of merely important to model”.)
>
> Relevant to the practicality issue mentioned later, we also think that our finding points to future research directions where robots attempt to influence their demonstrators to exhibit a specific bias in order to make them more informative. While it’s really hard for people to be pedagogical, it might be much easier for them to act myopically. Of course, how to do this well and whether it works out is an open question, but our paper uncovers that as a potentially new avenue for improving learning.
>
> >  it is important to discuss how this influences the contribution of this paper along with results from the above point demonstrating the effect of assuming wrong or incomplete irrationality when multiple irrationalities exist.
>
> As with reviewer 3 and 4, this reviewer also raises questions of practicality of our finding: since we usually don’t know the human’s bias, we can’t take advantage of their biased behavior being more informative. This is a valid point, but:
> * The finding itself has scientific value intrinsically, as it’s pointing out a surprising fact about biased behavior. Note that our paper is not a typical “here’s a new method” ML paper, it’s merely an analysis, one that arguably produced a surprising finding about a well-studied topic and therefore has value in itself.
> * The finding encourages taking biases seriously: not just because we make the wrong inference if we don’t, but because there is something to be gained by modeling them. It serves as an argument for further research in understanding and modeling enough about biases.
> * The finding also points to future research directions where robots attempt to influence their demonstrators to exhibit a specific bias (like myopia) in order to make them more informative. While it’s really hard for people to be pedagogical, it might be much easier for them to act myopically. Of course, how to do this well and whether it works out is an open question, but our paper uncovers that as a potentially new avenue for improving learning.

---

### Official Review · AnonReviewer4 · 2020-10-28
**interesting but impractical**

**Rating:** 4
**Confidence:** 3

**Review:**

##########################################################################

Summary:
This paper proposed modifications to the Bellman equation to capture known human irrationalities and showed that the reward under some conditions (the type of irrationality and parameter settings) can be better inferred compared to a rational agent. The authors demonstrated this through simulations in three different environments with different complexities and provided theoretical analyses to support this empirical finding. The authors further showed and discussed the effects on reward inference when the assumed parameter and assumed type of irrationality is misspecified.

##########################################################################

Reasons for score:
Overall, I vote for rejecting. The idea and results presenting are indeed very interesting and potentially promising, but I found it is to be impractical and based on lots of assumptions that easily fail. I elaborate on this below.

##########################################################################

Pros:

1. The writing is very clear and easy to follow.

2.  The paper nicely applies results from studies on human behavior to AI. The idea is very interesting and can potentially inspire lots of new works in this direction. It opens up more questions than it answers.

3. The authors showed the effects in three different environments, which demonstrates its generalizability to some degree. The authors also included theoretical results to support their empirical findings.

##########################################################################

Cons:

1. It seems that based on the results the author presented, we need to at least have prior knowledge about what type of irrationality humans exhibit (if not the exact parameter), this seems impractical to me and the authors did not discuss any potential approach to obtain this information. I think it is to some degree possible, as they were identified based on behavioral data empirically in human behavioral studies.

2. The authors also noted that humans exhibit a mix of irrationalities in 2.1, while the papers only considered cases where there’s a single irrationality.

3. In 2.1, the authors reasoned that it seems to them impossible to use actual human data, thus the paper is based on simulation. While simulations are valuable before diving into real data, I don’t think it “address these issues”. In addition, if what the authors described in 2.1 paragraph 1 are all true, it’s hard for me to see any value in this paper as it is not practical at all, i.e. the authors performed all analyses based on a miracle condition. Personally, I don’t think what authors claimed in paragraph 1 are all true, as there is a large body of experimental and computational studies in economics/psychology/neuroscience that try to answer those questions, and also noted by the authors, those irrationalities were experimentally observed.

4. The formulations of various irrationalities all look reasonable to me, but it would be better if the authors can validate that those formulations actually replicate human irrationalities in the same context as in the previous literature. Otherwise, it seems the choice of formulation kind of arbitrary to me.

#########################################################################

Other questions/suggestions:
In 2.3.3 Optimism/Pessimism, the first line after the equation, should it be s’ in V_i(s) instead of s? I supposed the optimism is formulated as the agent is expected to transit into a good state with higher probability, thus it should depend on the value function of the state it transits into.

I’m wondering how worse it does for reward inference if we assume rationality under different irrationalities. It would be great to add another curve in all plots in Fig. 3 to indicate the performance when ignoring irrationality. This result might validate that it is crucial to model irrationality.

For figure 4, the choice of beta = 10 seems kind of arbitrary to me. Is it the best performing parameter?

What are the parameters used in the simulation for figure 11 (the true parameter)? It would be good to put a vertical bar to indicate the case when the true and assumed parameters are the same. I supposed it is the value when the loss is minimal.

---

> ### Author Response · Authors · 2020-11-24
> **Response to Reviewer 4**
>
> Thanks for the detailed review!
> We’re happy that you found our paper easy to read, and also agree that the research direction is fruitful and of interest to the community.
>
> > It seems that based on the results the author presented, we need to at least have prior knowledge about what type of irrationality humans exhibit (if not the exact parameter), this seems impractical to me and the authors did not discuss any potential approach to obtain this information.
>
> As the reviewer says later in the review, psychologists and behavioral economists have identified several types of irrational behavior that humans exhibit. In addition, some recent work in machine learning has also looked at this problem (for example Shah et al 2019, which we cite).
>
> We do agree that our paper doesn’t discuss in detail the specific ways in which we could infer human irrationality. Instead, our work focuses on the effects of a known (through whatever method) or unknown bias on reward inference. We think that our analysis emphasizes why this is an important area of future work, and will add more discussion of this.
>
> >The authors also noted that humans exhibit a mix of irrationalities in 2.1, while the papers only considered cases where there’s a single irrationality.
>
> This is a good point. We agree that there is further work to be done involving mixes of irrationalities that could be added to the appendix. We expect our results to hold even for a mix of irrationalities. That being said, we feel that our paper makes an important contribution to understanding the effects of biases on reward learning without considering mixes of the different irrationalities we consider in our work.
>
> Regarding this point:
> > In 2.1, the authors reasoned that it seems to them impossible to use actual human data, thus the paper is based on simulation. While simulations are valuable before diving into real data, I don’t think it “address these issues”.…
>
> We agree that there are situations where researchers have experimentally observed biases. In some cases, the researchers use settings where the human subjects’ reward and biases are known and can be easily inferred mathematically, but then this is the same situation as our work. In other cases, the bias is only described heuristically, which means we can’t use it in our analysis without formalizing it with many researcher degrees of freedom. And in some cases the specific forms of the biases (or even their existence) are contested as the findings can be explained by different rewards. In addition, with human data, we won’t be able to go into the type of detail we did in this work. For example, it would be significantly harder to set the parameter of different biases like the horizon for myopia.
>
> We also agree that our arguments in 2.1 should not be construed to mean that we could never infer biases from human data. Instead, we see the arguments in 2.1 as explaining why we chose not to use human data in our analysis. And we also believe that our analysis could be used to motivate future work in this area.
>
> > ... In addition, if what the authors described in 2.1 paragraph 1 are all true, it’s hard for me to see any value in this paper as it is not practical at all ...
>
> As with reviewer 3, this reviewer raises an important question of practicality of this finding: since we usually don’t know the human’s bias, we can’t take advantage of their biased behavior being more informative. This is a valid point, but even if we can’t infer the bias and reward of humans today:
> * The finding encourages taking biases seriously: not just because we make the wrong inference if we don’t, but because there is something to be gained by modeling them. It serves as an argument for further research in understanding and modeling enough about biases.  As the reviewer points out, cognitive science, and behavioral economics have identified many biases. So while the approach in this work may not immediately scale, we believe the results can still inform research in this area.
> * The finding also points to future research directions where robots attempt to influence their demonstrators to exhibit a specific bias (like myopia) in order to make them more informative. While it’s really hard for people to be pedagogical, it might be much easier for them to act myopically. Of course, how to do this well and whether it works out is an open question, but our paper uncovers that as a potentially new avenue for improving learning.
>
>
> > ..., i.e. the authors performed all analyses based on a miracle condition
>
> In addition to the above points, we do perform a sensitivity analysis in section 5 (and the appendix).

---

> ### Author Response · Authors · 2020-11-24
> **Responses to the short questions at the end:**
>
> * Yes, it should be V_i(s’) - that is a typo on our part. We’ve corrected it.
> * Failing to model the irrationality type tends to lead to really bad results, as shown in figure 4. We excluded the lines from figure 3 as the performance isn’t on the same scale.
> * Beta = 10 was chosen because it was the best performing Beta for a Boltzmann-Rational human.
> * Yes, the parameters used are indicated by where the loss is minimal. However, this isn’t very clear for some of the biases, so we will add the specific parameters used into the text.

---

### Official Review · AnonReviewer3 · 2020-10-28
**Great start but not quite there yet**

**Rating:** 4
**Confidence:** 5

**Review:**

The authors consider the following disconnect:
1) Bellman equation based models make the implicit assumption that the agent making the decisions is rational in a very strong sense: they know the model correctly, the discount the future exponentially, they sum the (discounted) rewards as the total reward of a policy
2) People don't behave in this way

The disconnect between 1 and 2 is a big problem when we use inverse RL to try to learn the underlying reward functions from demonstrations since we might learn completely the wrong reward function. This can be bad for many reasons including counterfactual inference for what the person would do as well as inference for welfare (which the authors do not discuss, more on this later).

Good Parts
- The authors recognize a major problem in inverse RL from human demonstrations
- The authors have a list of known biases and how to parametrize them in reward inference, this list is larger than what is covered in past literature
- The authors show how badly inference can fail when the model is misspecified

Places to Improve
- The authors present their current contribution as more novel than it really is.
There is a huge literature in psychology and behavioral economics (examples: Ainslie 2001 is a whole book on the topic, O'Donoghue and Rabin 1999 American Economic Review, Laibson 1997 Quarterly Journal of Econ.) that specifically focus on what happens if someone is rational (here, discounts exponentially) but is actually irrational (discount hyperbolically).

The overall point here is both one of being able to predict behavior and one of welfare. Here, welfare refers to the following problem: if a rational agent chooses to e.g. smoke cigarettes, then clearly they prefer smoking to not smoking. On the other hand, a hyperbolic discounter may choose to smoke in the moment but also choose to take actions like throw cigarette packs away because they don't want to be tempted smoke in the future. In this case, we would infer that the smoking action was, in some sense, a mistake. Something, that we can never learn if we assume that agents are rational.

In addition, there is recent literature in the AI ethics community on precisely this same question (Peysakhovich 2019, AI Ethics and Society) that makes exactly the same point: if you do inverse RL assuming a rational model when individuals are irrational, you will learn exactly the wrong thing.

- It is unclear from the paper whether the authors' current approach is doable in any real situation
The current setup has the issue that when doing inference we must know exactly *the way* in which the actor was irrational in order to be able to get the gains from using the correct model. However, in practice we know that agents may have one of many biases, but don't know exactly which one they have. The current theorems tell us that "there exist MDPs"where we can tell which irrational model is correct, however it's not immediately clear what characteristics those MDPs have and whether, for example, just by having access to the MDP and a trajectory we can say that this trajectory is consistent or inconsistent with a rational model or a single type of irrationality.

The empirical counterpart to this is Figure 4 which compares irrational actors and inference with a rational vs the correct irrational model, however since the correct irrational model isn't ever known, it seems like the better comparison would be to show rational vs. all types of irrational and see whether the correct type of irrationality is picked up.

---

> ### Author Response · Authors · 2020-11-24
> **Response to Reviewer 3**
>
> Thanks for the detailed review! We are very happy that you agree that the problem of studying IRL for irrational demonstrators is an interesting problem, and also recognize the value in enumerating a large list of biases in a systematic way.
>
> We would also like to thank you for the pointers to more related work in the Behavioural Economics/Psychology field. We agree that there has been a large amount of work in studying the impacts of specific biases (or families of biases), as discussed in our introduction, but (as recognized by the reviewer) unlike prior we provide a systematic covering of multiple biases (of various different types) in the same setting.
>
> More importantly, we believe that another important finding from our work is not just that it is important to model biases, but that biases can actually be helpful for inference. That is, a biased demonstrator can be more informative than a rational demonstrator, if the bias is known. (See our top-level comment, “Biases can be more informative - instead of merely important to model”.)
>
> > The authors present their current contribution as more novel than it really is.
>
> We’re not aware of any prior work that claims that biases can be actually helpful for inference.
>
> > It is unclear from the paper whether the authors' current approach is doable in any real situation
>
> The reviewer raises an important question of practicality of this finding: since we usually don’t know the human’s bias, we can’t take advantage of their biased behavior being more informative. This is a valid point, but
> * The finding itself has scientific value intrinsically, as it’s pointing out a surprising fact about biased behavior. Note that our paper is not a typical “here’s a new method” ML paper, it’s merely an analysis, one that arguably produced a surprising finding about a well-studied topic and therefore has value in itself.
> * The finding encourages taking biases seriously: not just because we make the wrong inference if we don’t, but because there is something to be gained by modeling them. It serves as an argument for further research in understanding and modeling enough about biases.  As the reviewer points out, cognitive science, and behavioral economics have identified many biases. So while the approach in this work may not immediately scale, we believe the results can still inform research in this area.
> * The finding also points to future research directions where robots attempt to influence their demonstrators to exhibit a specific bias (like myopia) in order to make them more informative. While it’s really hard for people to be pedagogical, it might be much easier for them to act myopically. Of course, how to do this well and whether it works out is an open question, but our paper uncovers that as a potentially new avenue for improving learning.
>
> >The current setup has the issue that when doing inference we must know exactly the way in which the actor was irrational in order to be able to get the gains from using the correct model.
> > however since the correct irrational model isn't ever known, ...
>
> We do analyze this result in section 5 the paper. In Figure 11, we show that even when the inferer assumes the incorrect parameters for the irrationality, as long as they are not completely off base, the inferer still does better than assuming Boltzmann rationality.
>
> > if you do inverse RL assuming a rational model when individuals are irrational, you will learn exactly the wrong thing.
>
> Again, one important finding from our work is not just that it is important to model biases, but that biases can actually be helpful for inference. (See our top-level comment, “Biases can be more informative - instead of merely important to model”.)

---

### Author Response · Authors · 2020-11-24
**Biases can be informative - instead of merely important to model.**

We want to first and foremost clarify that we think that one of the major findings is that biases can make behavior more informative. Importantly, we are not (just) saying the trivial thing that if the human has a bias, it’s better to know about it than to assume rationality. We are saying something much more surprising: that the human having a bias in the first place is better than them being rational.

For example, a human driver optimizing for a short horizon ends up revealing much more about their preferences than a rational human driver, optimizing for the full horizon.

Why is this interesting? It is tempting to assume that biases make the behavior more opaque, and we’re finding the opposite. The most related finding to ours is that of pedagogical/legible behavior: that teachers trying to be informative will deviate from optimal behavior. But legible behavior is (in a sense) supra-rational: it requires the demonstrator to be modeling the learner and optimizing for their understanding. In contrast, biases like myopia are sub-rational, they're doing something. It is therefore somewhat surprising that despite polluting the optimality of the behavior, they in fact make it more informative without the demonstrator purposefully trying to be informative.

We just wanted to make sure this finding came across -- again, it’s not that it’s better to assume biases if they are there, instead it’s that it’s in a sense fortunate for biases to be there in the first place. We’re concerned that this didn’t come across in the paper, and would really appreciate some advice on how to emphasize it and explain it better.

For example, Reviewer 1 claims “This work only demonstrates that (in contrast to the findings of prior work of Shah et al.) knowing the true irrationality model outperforms assuming Boltzmann-rationality by a lot”. However, in our work we actually demonstrated that the presence of the bias (when known) improves reward inference, not merely that correctly knowing the bias helps. Reviewer 3 characterized our results as showing “how badly inference can fail when the model is misspecified”. However, we also showed result that biases are helpful for inference.

---

### Decision · Program_Chairs · 2021-01-07
**Final Decision**

**Decision:**

Reject

**Comment:**

Two very confident and fairly confident reviewers rate this paper ok but not good enough, and two other fairly confident reviewers rate the article below the acceptance threshold. Therefore I must reject the article. The reviewers provided encouraging comments and suggestions on how the manuscript could be improved, which I hope the authors will find useful.